# Time-Dependent Seismic Performance of Coastal Bridges Reinforced with Hybrid FRP and Steel Bars

**DOI:** 10.3390/ma15155293

**Published:** 2022-08-01

**Authors:** Wei Yuan, Zhong-Kui Cai, Xiaolan Pan, Jun Lin

**Affiliations:** 1School of Civil Engineering and Architecture, Jiangsu Open University, Nanjing 210036, China; 13b933008@hit.edu.cn (W.Y.); aslinjun@163.com (J.L.); 2College of Civil Engineering, Nanjing Tech University, Nanjing 211816, China; 3College of Civil Engineering, Taiyuan University of Technology, Taiyuan 030024, China; panxiaolan@tyut.edu.cn

**Keywords:** hybrid reinforced concrete bridges, fiber-reinforced polymer bars, marine environment, time-dependent seismic performance, hysteretic analysis, dynamic time-history analysis

## Abstract

To increase the durability and seismic resilience of coastal bridges, a hybrid reinforced concrete (HRC) bridge that incorporates both glass fiber-reinforced polymer (GFRP) bars and steel bars is proposed. The time-dependent seismic performance of the HRC bridge is comprehensively investigated at three levels, namely the material, bridge column and bridge structure levels. First, the decrease of tensile strength of GFRP bars over time is analyzed based on the Arrhenius theory, and corrosion initiation time and performance deterioration of steel bars are determined by Fick’s second law and an empirical formula. Second, an efficient finite element modeling method for aging HRC bridge columns is proposed. Simulation of the compression/tension behavior and the fracture failure of the GFRP bar is described. Hysteretic analysis is further conducted to investigate the time-dependent energy dissipation, ductility, residual displacement, bearing capacity and post-yield stiffness ratio. Meanwhile, comparisons of HRC bridge columns to reinforced concrete (RC) references are provided. Third, the seismic demand and damage evolution of deteriorated HRC bridge structures are investigated through dynamic time-history analysis. The results indicate that the corrosion-resistant GFRP bars contribute to improving the bearing capacity and to reducing the residual displacement of the HRC bridge. With an increase in service time, the seismic damage to the bridge column, abutment and expansion bearing increases, but the damage to fixed bearing decreases. Research results presented herein show that the HRC bridge is a promising alternative structure scheme in the marine environment.

## 1. Introduction

Due to abundant salt and high humidity, structures in marine environments are vulnerable to corrosion damage. For conventional reinforced concrete (RC) bridges, continuous penetration of chloride ions will induce strength degradation and diameter reduction of steel bars [1], decrease of the bond capacity [2,3] and cracking of the concrete [4]. In seismic-prone regions, the corrosion damage would significantly reduce structural load-resisting capacity and ductility. 

Composite materials made of fibers incorporated in a polymeric resin, namely the fiber-reinforced polymer (FRP), have advantages of both high corrosion resistance and large tensile strength [5]. Therefore, FRP bars have emerged as a promising alternative to steel bars in coastal structures. A hybrid reinforced concrete (HRC) bridge that employs both glass FRP (GFRP) bars and steel bars is proposed and studied in this paper. Understanding the time-dependent seismic performance of such HRC bridges is an important issue for engineers and researchers to assure its safety and determine maintenance strategies.

Plenty of efforts have been devoted to the corrosion-resistant property of the FRP bars [6,7,8]. Aggressive solutions, such as saline, acid and alkaline solutions are commonly adopted to simulate the corrosive environment. The deterioration behavior is generally described by a decrease of tensile properties, and sometimes supplemented by a change of elastic modulus and compression strength. Al-Salloum et al. [9] conducted an experimental study to investigate the properties of new generation of GFRP bars exposed to ten environmental conditions for 6, 12 and 18 months. The results demonstrated that the tensile modulus of elasticity exhibited negligible loss regardless of the surrounding environment and exposure period, and bars in tap water and alkaline solution at 50 °C generated the most severe damage to the tensile strength. Wu et al. [10] assessed the residual tensile strength of stressed and unstressed basalt FRP (BFRP) bars after being subjected to four types of simulated harsh environments at 25, 40, and 55 °C with duration times of 21, 42 and 63 days. The effect of the alkaline solution on the durability of BFRP bars was proved to be most notable. Furthermore, a sustained stress larger than 20% of the ultimate strength was discovered to have the effect of accelerating the degradation process. 

The tests described above were all adopting the direct immersion method, which would overestimate the deterioration degree. To obtain more reasonable results in the actual service environment, some corrosion tests have been carried out on the FRP bars embedded in the concrete [11,12]. The tensile strength, bond capacity and microscopic phenomenon have been studied. 

Focusing on the time-dependent tensile strength of FRP bars in the whole service life, several prediction models have been proposed. Bank et al. [13] pointed out that the tensile strength retention was proportional to the logarithm of service time. Beddows et al. [14] presented a static fatigue model to reveal the relationship between strength and time. Phani et al. [15] assumed that the attenuation mechanism of FRP bar was peel failure and proposed a model containing the parameters of characteristic and service time. All the models were obtained based on the Arrhenius equation [16].

Aside from the corrosion-resistant property, the high-strength feature of the FRP bar further promotes its investigations and applications. Hales et al. [17] tested short and slender columns reinforced with both GFRP and steel longitudinal bars. Axial and eccentric load-carrying capacities of such HRC columns were studied. Research indicated that GFRP bars were a promising method of reinforcement for columns with large eccentricities. Sun et al. [18] conducted cyclic loading tests on four columns with different combinations of FRP bars, steel bars and steel-FRP composite bars. The strain distribution, plastic deformation, post-yield stiffness, displacement ductility and equivalent viscous damping coefficients were analyzed. Yuan et al. [19] carried out cyclic tests on hybrid FRP and steel reinforced engineered cementitious composite columns. Test results indicated that the HRC columns exhibited higher self-centering capabilities than traditional RC columns. Chellapandian et al. [20] repaired severely damaged RC columns with carbon FRP (CFRP) bars and external wraps, and the strengthened columns became HRC members. Experiment results revealed that such a strengthening method can effectively restore the eccentric load-carrying capacity of the column. Furthermore, the first two authors of this paper investigated HRC precast segmental bridge columns which reinforced with CFRP bars and steel bars [21]. Quasi-static test results demonstrated that CFRP bars contributed to decreasing residual displacements of precast bridge columns. In addition, much effort has been put into the exploration of the flexural behavior of the HRC beams [22,23].

Research regarding the issue of seismic performance of HRC bridges in the marine environment over the whole life cycle, however, has seldom been involved in previous studies. To address this problem, this paper performed a hysteresis analysis and dynamic time history analysis on deteriorated HRC bridges. Three parts of content are organized as follows. (1) At the research level of materials: the deteriorations of the steel bars and FRP bars due to chloride corrosion are calculated based on the finite difference method, an empirical formula and Arrhenius theory. (2) At the research level of columns: the finite element modeling method of aging HRC bridge columns is proposed and verified. Time-dependent hysteretic behavior of HRC columns is analyzed. (3) At the research level of structures: the seismic demand and damage evolution of different components of the coastal HRC bridges over time are evaluated through dynamic time history analysis. The content of materials is the foundation of the study. Based on the deterioration results of steel bars and GFRP bars, the time-dependent degradation models of the coastal bridge columns can be determined. As the bridge columns are the main lateral-load resisting members of the bridge, the content of columns is the key and basis for the content of structure. The geometric dimensions and physical properties of the sound and aging bridge columns are the same as the ones in coastal bridges. Briefly, the responses of columns are used for analysis of structures, while the responses of materials are used for analysis of columns.

The main contribution of this study is summarized as follows: (1) the simultaneous deterioration of GFRP bars and steel bars over time is considered in the analysis, thus to develop a comprehensive understanding of the chloride induced material corrosion; (2) the time-dependent seismic performance of the coastal HRC bridges during the whole life cycle is investigated, thus to provide an alternative to traditional RC bridges in the marine environment. The preceding two aspects have seldom been studied previously.

## 2. Material Deterioration Models

In the marine environment, the different corrosive characteristics in atmospheric, splash, tidal and submerged zones result in significant differences in time-dependent performance of the bars. In this paper, the research focuses on corrosion resulting from sea-salt particles floating in the air and assumes that the diffusion process is the dominant mode of chloride intrusion. In addition, chloride-induced corrosion is mainly considered in the deterioration of the steel bars and FRP bars in the HRC bridge columns.

### 2.1. Steel Bar Deterioration

The deterioration process of the steel bar can be divided into two stages, i.e., the constant performance stage and the continuous deterioration stage, respectively before and after the corrosion is initiated. The corrosion initiation time is defined as the moment when the total chloride concentration near the steel bars reaches the threshold concentration. Fick’s second law is derived from the first law and the mass conservation equation, which can predict the concentration over time, and has been widely used in the diffusion analysis [24,25,26]. Based on Fick’s second law, the process of chloride penetration into concrete can be described as [27]
(1)$$\frac{\partial C_{f}}{\partial t}=\frac{\partial }{\partial x}\left(D_{\mathrm{cl}}^{*}\frac{\partial C_{f}}{\partial x}\right)$$
where x is the depth measured from the concrete surface; t is the service time; $C_{f}$ is the free chloride concentration at the location x over time (t); $D_{\mathrm{cl}}^{*}$ is the apparent diffusion coefficient and can be given as
(2)$$D_{\mathrm{cl}}^{*}=D_{\mathrm{cl}}/\left(1+\frac{1}{\omega_{e}}\frac{\partial C_{b}}{\partial C_{f}}\right)$$
with
(3)$$D_{\mathrm{cl}}=D_{\mathrm{cl},\mathrm{ref}}F_{1}\left(h\right)F_{2}\left(T\right)F_{3}\left(t\right)F_{4}\left(C_{f}\right)$$
where $D_{\mathrm{cl}}^{}$ and $D_{\mathrm{cl},\mathrm{ref}}^{}$ is the effective and reference diffusion coefficient, respectively; $F_{1}\left(h\right)$, $F_{2}\left(T\right)$, $F_{3}\left(t\right)$ and $F_{4}\left(C_{f}\right)$ are the influence coefficients that are determined by the relative humidity, ambient temperature, concrete age and free chloride concentration, respectively; $\omega_{e}$ is the volume of evaporable water per volume of concrete; $\partial C_{b}/\partial C_{f}$ is the binding capacity describing the relationship between the free and bound chloride ions in concrete at a constant temperature. According to Freundlich isotherm [28], it is expressed as
(4)$$\partial C_{b}/\partial C_{f}=\alpha_{F}\beta_{F}C_{f}^{\beta_{F}-1}$$
where $\alpha_{F}$ and $\beta_{F}$ are the binding constants and equals to 1.05 and 0.36, respectively; the relationship among the total ($C_{t}$), bound ($C_{b}$) and free ($C_{f}$) chloride concentration is as follows
(5)$$C_{t}=C_{b}+\omega_{e}C_{f}$$

A review of models proposed in the literature [28,29,30] is performed and the mathematical descriptions of $F_{1}\left(h\right)$, $F_{2}\left(T\right)$, $F_{3}\left(t\right)$ and $F_{4}\left(C_{f}\right)$ are summarized in Table 1. According to the research findings obtained by Bažant et al. [31], Saetta et al. [32] and Xi et al. [30], the assigned values for the parameters in the functions are also listed in Table 1. 

Based on the multi-physical field simulation software (COMSOL Multiphysics), Equation (1) is solved. The time-dependent total chloride ions concentration near the steel bars is obtained. In the simulation, the free chloride content in the concrete at the initial conditions is assumed to be zero. With accumulation rates of 0.10 % concrete weight per year [33], the surface chloride content increases continuously in the first 10 years and then remains constant. The threshold total chloride content is taken as 1.2 kg/m^3^ [33]. Considering the environmental data of the bridge investigated in the study, the average relative humidity and ambient temperature is adopted as 0.731 and 290.468 ${}^{{}^{\circ}}K$, respectively. For the longitudinal steel bar and stirrup in the bridge column, the cover depths are 50 and 40 mm, respectively. 

Figure 1 shows the simulation results of the distribution of total chloride content in the bridge column after servicing for 50 years, obtained by COMSOL software. As seen from the figure, a larger distance to the concrete surface would result in smaller chloride accumulation. Figure 2 shows the time-dependent total chloride content at the surface of the steel bars. By comparing with the threshold value, the corrosion initiation time of the longitudinal steel bar and stirrup is obtained to be 15.8 and 7.5 years, respectively.

After the corrosion is initiated, the steel bars exhibit continuous performance deterioration in the service life, including a decrease in the diameter and yield strength. To estimate the residual diameter D(t) after servicing *t* years, a simple and effective model is chosen and expressed as follows [34]:(6)$$D\left(t\right)=\left\{ \begin{array}{l} D_{0}\\ D_{0}-0.023i_{\mathrm{corr}}\Delta t \end{array}\right.$$
where $D_{0}$ is the initial diameter of the steel bar; $i_{\mathrm{corr}}$ is the corrosion current density; Δt is the time elapsed since the corrosion is initiated.

The residual yield strength $f_{y}\left(t\right)$ is given as [35]:(7)$$f_{y}\left(t\right)=\left[1-0.005Q_{\mathrm{corr}}\left(t\right)\right]f_{y0}$$
with
(8)$$Q_{\mathrm{corr}}\left(t\right)=\left[1-{\left(D\left(t\right)/D_{0}\right)}^{2}\right]\times 100$$
where $Q_{\mathrm{corr}}\left(t\right)$ is the mass loss rate of the steel bars; $f_{y0}$ is the initial yield strength.

A range of 0.5 $\mu A/{\mathrm{cm}}^{2}$ to 5 $\mu A/{\mathrm{cm}}^{2}$ has been suggested for the corrosion current density when the exposure class of the bridge is airborne seawater [36]. In this study, the value of 4 $\mu A/{\mathrm{cm}}^{2}$ is adopted. The initial diameter of the longitudinal steel bar and stirrup is 25 and 10 mm, respectively. The initial yield strengths are both 452 MPa. Based on Equations (6)–(8), Figure 3 gives the estimated results of performance deterioration of the steel bars due to chloride corrosion. As shown in the figure, the diameter decrease rates in the whole life cycle for the longitudinal steel bar and stirrup are 31.0% and 85.1%, respectively. The corresponding yield strength shows decrease rates of 26.2% and 48.9%, respectively.

### 2.2. FRP Bar Deterioration

Compared to the conventional steel bars, the FRP bars exhibit higher tensile strength and better durability. Therefore, FRP bars are also designed in the coastal bridge column. Various accelerated durability tests [6,7,8] revealed that the degradation of the FRP bars in alkaline, acid, and saline solution is quite different. To accurately predict the long-term behavior of the FRP bars in the marine environment, the deterioration of the FRP bars used as internal reinforcement of concrete subjected to a saline solution should be assessed. In addition, the influence of resin matrix type, fiber type and fiber content on the performance of the FRP bar should be considered.

The research on the durability of FRP is mainly based on experiments and there are no widely accepted recommended values. Hence, the test data obtained by Robert et al. [11], which satisfied the above requirements, are adopted, and the extended analysis is performed. In the test, the GFRP bars were used and had a nominal diameter of 12.7 mm. The average tensile modulus and mean tensile strength were 46,300 MPa and 786 MPa, respectively. The concrete specimens with GFRP bars embedded in the middle were completely immersed at three different temperatures ($23\;{}^{{}^{\circ}}C$, $40\;{}^{{}^{\circ}}C$ and $50\;{}^{{}^{\circ}}C$) and were removed from the saline solution after four different periods of time (60, 120, 210 and 365 days). For each corrosion condition, six GFRP bars were tested.

The residual tensile strength is considered the primary indicator for the GFRP bars to reflect the performance degradation due to chloride-induced corrosion. To obtain the time-dependent residual tensile strength over the whole service life, the test data are processed according to the Arrhenius model [16]. The Arrhenius model is often used in the field of accelerated aging of composites and can reveal the relationship of degradation of mechanical properties in bars at different temperatures. It can predict the material property under a natural environment through the data obtained from short-term artificial acceleration tests and has been widely adopted in long-term performance prediction of the FRP materials [37,38]. The model can be given as
(9)$$k=A\exp\left(\frac{-E_{a}}{RT}\right)$$
where *k* is the deterioration rate; *A* is the constant influenced by the material and deterioration process; $E_{a}$ is the activation energy; *R* is the gas constant; *T* is the thermodynamic temperature. The Arrhenius model assumes only one deterioration mechanism operates during the reaction and will not change with time and temperature. Equation (9) can be transformed into
(10)$$\frac{1}{k}=\frac{1}{A}\exp\left(\frac{E_{a}}{RT}\right)$$
with
(11)1/k=t
where *t* is the service time. By taking the logarithm, Equation (10) is rewritten as
(12)$$\ln\left(\frac{1}{k}\right)=\frac{E_{a}}{R}\frac{1}{T}-\ln A$$

Based on the test data and Equations (11) and (12), the estimated results of Arrhenius curves are given in Figure 4a. The X-axis and Y-axis represent the corrosion temperature and tensile strength retention, respectively. The ten curves represent data with different service times. For the bridge investigated in the paper, the average temperature of the surrounding marine environment is assumed to be $20\;{}^{{}^{\circ}}C$. Thus, the inverse temperature is calculated to be 3.41. The time-dependent tensile strength retention of the GFRP bar in the whole life cycle can be obtained from the figure and exhibits a decreasing trend. Compared to the initial value of 1.0, the tensile strength retention is decreased to 0.77 after servicing for 100 years. The estimated result of residual tensile strength calculated by initial strength multiplying strength retention is shown in Figure 4b.

## 3. HRC Bridge Column Assessment

As the main lateral force resisting member of the bridge, the HRC bridge column is assessed at first. Based on the material deterioration models discussed above, the time-dependent seismic performance analysis is conducted here. 

### 3.1. Time-Dependent Finite Element Models

With a time interval of 10 years, a series of finite element models of the HRC bridge columns are built using the open-source finite element analysis software OpenSees [39]. In the models, the diameter and height of the bridge column are 1.0 and 6.5 m, respectively, resulting in a shear span ratio of 6.5. Eighteen 25 mm longitudinal steel bars and eighteen 12.7 mm GFRP bars are designed as hybrid longitudinal reinforcements. Furthermore, 10 mm circular stirrups with a space of 100 mm are set for the sound structure. The time-dependent diameter and mechanical properties of the bars are consistent with the results in Section 2. The compressive strength of the concrete is 35.4 MPa. The gravity load applied to the bridge column is 2779 KN with the corresponding axial compression ratio of 0.1. 

The finite element modeling method is described in Figure 5. Part I shows the schematic diagram of the HRC bridge column. Part II exhibits the elements adopted in the simulation. It should be noted that, not only is the Nonlinear Beam-Column Element used to consider the spread of plasticity along the column, the Zero-Length Section Element is also adopted to capture the strain penetration effect at the column-to-footing intersection. The *P* in part II equals the gravity load of 2779 KN, while the *F* is constantly changing as the displacement control is applied in the cyclic loading process. Part III shows the fiber section division and the material models of the column. The Concrete02 is used for the cover and core concrete fiber, and the ReinforcingSteel is used for the steel bar fiber. A uniaxial engineered cementitious composite material model ECC01 is considered for the simulation of GFRP bar fiber. Parts IV, V and VI give the constitutive relationships of concrete, steel bar and GFRP bar, respectively. 

As the difficult point in the simulation, the stress-strain relationship of the GFRP bar is discussed in detail here. In the literature [21,40], it had been verified that the GFRP bars may fail in tension or in compression. Furthermore, a fracture phenomenon could be observed. To accurately simulate the behavior of the GFRP bar, meanwhile, improving computational stability, two aspects are considered in the material model as follows

(1) The compression behavior of the GFRP bar is contained in the material model, including the mechanical properties and the possible fracture failure. The $f_{\mathrm{FRP}}^{\prime }$ and $E_{\mathrm{fc}}$ in Part VI are the peak strength and elastic modulus in the compression direction, respectively. Although research of $f_{\mathrm{FRP}}^{\prime }$ and $E_{\mathrm{fc}}$ has received much attention [41,42,43], no consistent conclusion is reached yet. In the previous tests conducted by the authors [44], $f_{\mathrm{FRP}}^{\prime }$ approximates 50% of the peak tensile strength $f_{\mathrm{FRP}}$ when the confinement effect of the surrounding concrete is accounted for. According to the ACI 440.1R-15 design code [5] and related research [43,45], $E_{\mathrm{fc}}$ can be appropriately described as
(13)$$E_{\mathrm{fc}}=(0.8\sim 1.0)E_{\mathrm{ft}}$$
where $E_{\mathrm{ft}}$ is the elastic modulus of the GFRP bar in the tensile direction. In this paper, $E_{\mathrm{fc}}=E_{\mathrm{ft}}$ is adopted.

(2) Instead of the immediate drop of the strength when the GFRP bar fails, a steep degradation is employed. As shown in part VI, the peak ($\varepsilon_{\mathrm{ftp}}$) and ultimate ($\varepsilon_{\mathrm{ftu}}$) tension strain is not identical, the same happens to the peak ($\varepsilon_{\mathrm{fcp}}$) and ultimate ($\varepsilon_{\mathrm{fcu}}$) compression strain. The relationship of the above four parameters is given as
(14)$$\varepsilon_{\mathrm{ftu}}=\alpha \varepsilon_{\mathrm{ftp}}$$
(15)$$\varepsilon_{\mathrm{fcu}}=\alpha \varepsilon_{\mathrm{fcp}}$$
where α is the amplification factor with a value larger than 1.0. A trial procedure in this study indicated that α = 1.0 and 1.2 generally yield the same column response, but the latter involves much less computational effort. As a result, α is taken as 1.2 in this paper. Similar modification in material modeling has also been implemented by Kwan et al. [46].

### 3.2. Model Validation

To guarantee the same seismic response between the tests and simulation, accuracy in simulating the time-dependent corrosion influence and FRP bar fracture phenomenon is quite important. As limited research has been conducted on the corroded HRC bridge column, the modeling method in this paper is verified through two steps. Firstly, compare the simulation and test results of the corroded RC bridge columns. Secondly, compare the two results of the sound HRC bridge columns. 

Based on the cyclic loading tests conducted by the authors previously [47], the first step is performed and exhibited in Figure 6. The D60 and D150 in the figure refer to the test specimens with accelerated corrosion time of 60 and 150 days, respectively. For each specimen, the constant axial load was 60 t and the corresponding axial compression ratio was 0.13. The lateral cyclic loads were imposed via the MTS actuator. The displacement amplitudes of 2 mm and 4 mm were both imposed for one cycle at first as the preloading displacements. Then, the displacement amplitudes were all repeated twice and increased as Δ_1_ = 8 mm, 2Δ_1_, 3Δ_1_, …, until the column failed. The loading speed was 0.2 mm/s, and the damage observation was conducted after each hysteresis cycle was completed. A stabilized response of the material was sustained in the loading process.

Table 2 also lists the comparison results of the energy dissipation ED, displacement ductility coefficient μ and secant stiffness at peak lateral load SS. In the table, the data at the left and right side of the slash are the test and simulation result, respectively. The error (%) is calculated and put in the parentheses. The SF indicates structural failure is earlier than 104 mm, thus no test data are obtained. It is seen that among the 24 comparisons, 15 results are with an error of less than 10%. 

The second step of the verification is performed based on the tests conducted by Ibrahim et al. [48]. Comparisons for specimens ES-D8-J and IR-D10-J are given in Figure 7. Among them, the specimen ES-D8-J was with six 8 mm diameter FRP bars placed external to the steel stirrups, and the specimen IR-D10-J was reinforced with four 10 mm diameter FRP bars placed at the same place as the longitudinal steel bars. The two columns were both subjected to a constant axial load of 40 kN and several excursions of lateral cyclic loading using a dynamic actuator with a capacity of 700 kN. The yield displacement Δ_0_ was determined to be 5 mm. The lateral loading sequence of specimens ES-D8-J and IR-D10-J started with two cycles of 0.5Δ_0_, followed by two cycles of Δ_0_, and then three cycles each of 2Δ_0_, 3Δ_0_, 4Δ_0_, 6Δ_0_, 8Δ_0_…, until the columns failed. All the failure phenomena during the loading of the two columns were observed carefully and notes were taken. The hysteresis cycles correspond to the stabilized response of the material. Figure 7 shows that finite element modeling results compare favorably with test results. 

### 3.3. Seismic Performance Analysis

Based on the finite element models of the sound and aging HRC bridge columns, structural time-dependent seismic performance is investigated every 10 years. To analyze the influence of FRP bars on the seismic behavior, the RC bridge column only reinforced with eighteen *ϕ*25 mm longitudinal steel bars and *ϕ*10 mm stirrups is also being assessed. It should be noted that the damage aggregation and deterioration of the coastal structures in the long-term service period is a complex process. Chloride corrosion, fatigue damage, creep phenomenon, carbonization and seismic phenomena can all influence the seismic performance of the coastal bridges, and all the factors should be considered in the analysis. However, it will make the problem quite complicated. To focus on the research, chloride-induced corrosion is chosen as the deterioration factor in the study.

The lateral load-top displacement curve can reflect the deformation characteristics, stiffness degradation and energy dissipation of the structure in the process of cyclic loading. Meanwhile, it is the basis for determining the restoring force model and performing the nonlinear seismic response analysis. Therefore, it is investigated first. Figure 8a shows the hysteresis curves of the HRC and RC bridge column when the structures are just exposed to the marine environment. Figure 8b illustrates the hysteresis curves comparison of the HRC bridge column between the two service times of 0 and 100 years. The maximum loading displacements of 0 and 100 years HRC bridge column are determined based on two considerations. Firstly, the same loading protocol is chosen for a clear comparison; secondly, a large displacement amplitude is preferred to clearly reflect the influence of corrosion on the hysteretic curve of such components. The ascending and descending phase, and the steep drop phenomenon due to GFRP bar rupture (if it occurs) should be observed. As observed from the figure, the addition of the GFRP bars increases the peak load and slows down the descending of the lateral load after the peak point. Furthermore, the deterioration of steel and GFRP bars induced by chloride corrosion results in an obvious pinch phenomenon.

The residual displacement $\Delta_{\mathrm{Res}}$ obtained from the cyclic loading test refers to the displacement when the lateral load decreases to zero. Considering the possible asymmetry of the hysteresis curves induced by the material properties, $\Delta_{\mathrm{Res}}$ is given as
(16)$$\Delta_{\mathrm{Res}}=\frac{\Delta_{\mathrm{Res}}^{+}+\Delta_{\mathrm{Res}}^{-}}{2}$$
where $\Delta_{\mathrm{Res}}^{+}$ and $\Delta_{\mathrm{Res}}^{-}$ are the residual displacement at the positive and negative direction, respectively. Equation (16) is described in Figure 9a, meanwhile, the comparison results of $\Delta_{\mathrm{Res}}$ for the HRC bridge column at the service times of 0, 20, 40, 60, 80 and 100 years are shown. Figure 9b exhibits the residual displacements of the HRC and RC bridge column at the service times of 10, 50 and 90 years. As seen from Figure 9a, the corrosion of the GFRP and steel bars leads to a continuous decrease of the $\Delta_{\mathrm{Res}}$, whereas, a slight increase is observed in the first 20 years. This phenomenon is caused by the fact that the deterioration of the GFRP bars in this period is much more severe than the steel bars. Figure 9b indicates that the residual displacements of the RC bridge column are much larger than that of the HRC bridge column.

The energy dissipations of the bridge columns are analyzed and given in Figure 10. As seen in Figure 10a, a continuous decrease of the energy dissipation is generated with the increase in service time. Compared to the sound state, the energy dissipation of the HRC bridge column at 100 years decreases from 973 to 540 kN·m when the displacement amplitude is 260 mm, with the reduction rate of 44.5%. Figure 9b indicates that the same decrease trend is observed for the RC bridge column. In addition, a small difference in energy dissipation capacity is observed for these two types of columns.

Connecting the peak point of each level hysteretic curve under cyclic loading, the envelope curve is obtained. For the HRC bridge column, the time-dependent envelope curves are exhibited in Figure 11a. The solid circle in the figure refers to the occurrence of FRP bar fracture. It is seen that the fracture phenomenon does not occur in the loading process at 0 years. However, the increase in service time results in the earlier fracture of FRP bars. At the service times of 20, 40, 60, 80 and 100 years, the corresponding fracture top displacement is 240, 225, 220, 210 and 200, respectively. Figure 11b shows the comparisons of envelope curves between the HRC and RC bridge column. It is observed that the load capacity of the HRC bridge column is much higher, whereas a sudden decrease is generated when FRP bars fracture, and finally approximately the same with the RC bridge column.

The definition of the characteristic parameters of envelope curves is described in Figure 12. When the longitudinal steel bar yields, the yield displacement $\Delta_{y}$ and load $P_{y}$ are obtained. Peak displacement $\Delta_{p}$ and load $P_{p}$ correspond to the peak point of the envelope curve. The determination of the ultimate parameters is more complicated. As shown in Figure 12a, three possible failure phenomena can occur and, once one is reached, the ultimate state is reached and the ultimate displacement $\Delta_{u}$ and load $P_{u}$ are obtained. The $\varepsilon_{c}$ and $\varepsilon_{c,\mathrm{ls}}$ refer to the concrete compression strain and ultimate compression strain, respectively. According to Priestley et al. [49], $\varepsilon_{c,\mathrm{ls}}$ can be expressed as
(17)$$\varepsilon_{c,\mathrm{ls}}=0.004+1.4\frac{\rho_{v}f_{\mathrm{yh}}\varepsilon_{\mathrm{su}}}{f_{\mathrm{cc}}^{\prime }}$$
where $\rho_{v}$ is the stirrup ratio; $f_{\mathrm{yh}}$ and $\varepsilon_{\mathrm{su}}$ are the yield strength and ultimate strain of the stirrups, respectively; $f_{\mathrm{cc}}^{\prime }$ is the compressive strength of confined concrete. The displacement ductility coefficient μ is defined as $\Delta_{u}/\Delta_{y}$. Figure 12b illustrates the definition of the post-yield stiffness ratio K. It can reflect the relationship between post-yield stiffness $K_{2}$ and pre-yield stiffness $K_{1}$.

Table 3 summarizes the time-dependent results of the characteristic parameters. It can be concluded that the post-yield stiffness ratio K is gradually increased with an increase in service time, exhibiting the opposite trend to the other seven parameters. This phenomenon is due to the larger and larger hybrid ratio (area of GFRP bars/area of steel bars). During the whole service life, the increase rate for K and decrease rate for $\mu_{\Delta }$ are 37.9% and 31.7%, respectively.

## 4. HRC Bridge Assessment

### 4.1. HRC Bridge Model

A dynamic time-history analysis is performed in this section to gain a comprehensive understanding of the time-dependent seismic performance of the aging HRC bridges. The schematic diagram and finite element model of the bridge are shown in Figure 13 and Figure 14, respectively.

As shown in the figures, a four-span continuous steel girder bridge is investigated. The span length and deck width are 20 m and 9 m, respectively. Three elastomeric fixed bearings are resting on each bent beam, while the same number of elastomeric expansion bearings are set on each abutment. The HRC columns in the bridge are identical to the ones analyzed in Section 3 and are respectively tied to an individual pile foundation system. The gap presented between the deck and the abutment is measured to be 45 mm. 

The bridge is designed to have a service life of 100 years. Including the sound bridge model, 11 finite element models are built with a time interval of 10 years by the OpenSees software [39]. The corrosion-induced deterioration is simulated according to the description in Section 2. In the models, the element elasticBeamColumn is adopted for the superstructure as a linearly elastic response under seismic loading is expected. The element zeroLength is used in simulating the soil-abutment interaction, expansion bearing, fixed bearing and the pile foundation system. The element nonlinearBeamColumn is applied to the HRC column to consider the spread of plasticity. The concrete, steel bars and GFRP bars in the bridge columns have the same properties as the ones adopted in Section 3. The elastomeric pad and steel dowels are contained in the bearings. Between them, the former is considered with an initial stiffness of 3.35 kN/mm and a post-yield stiffness of 0 kN/mm and can be modeled using the Steel01 material. The latter is with an initial stiffness of 92 kN/mm and a post-yield stiffness of 1 kN/mm, and can be modeled by the Hysteretic material [50]. Furthermore, the linear and nonlinear springs with a length of zero are created between the column bottom and the assumed ground fixed point to reflect the translational and rotational behavior of the foundation. The same principle is applied in the simulation of soil-abutment and pile-abutment interaction. For simplification, the parameters adopted in the simulation are deterministic, i.e., the uncertainty and randomness are not taken into consideration. Furthermore, the shear deformation of the bridge columns is not included in the analysis as such components with a large shear-span ratio and the flexural response always dominates.

### 4.2. Ground Motion Selection

To investigate the evolutionary process of the seismic demand and damage of the HRC bridge, ten acceleration time histories of the ground motions are selected and listed in Table 4. The ground motion records are satisfied with the elastic response spectrum of the ground type D in Eurocode 8 and are applied to the structure in the longitudinal direction. Figure 15 shows the individual and median acceleration response spectrum. For the sound HRC bridge, the first natural period in the longitudinal direction is calculated to be 0.56 s, and the corresponding spectral accelerations ranged from 0.412 to 1.321 g.

### 4.3. Seismic Demand and Damage Evolution

For each HRC bridge model, ten nonlinear dynamic analyses are conducted. The seismic demand and damage evolution of different components are discussed. Due to limited previous research on the seismic performance of the HRC bridges in the marine environment, the comparison of the simulation results to the test results is not performed in this part. In future research, the authors will conduct the tests concerning about time-dependent seismic performance of the coastal HRC bridges for further investigation and verification.

Focusing on the HRC column, the drift ratio and curvature ductility demand ratio ($R_{\mathrm{cdd}}$) that can reflect its damage degree are firstly investigated. The $R_{\mathrm{cdd}}$ that can quantify the deformation degree of the cross-section under seismic excitation is expressed as
(18)$$R_{\mathrm{cdd}}=\frac{\varphi }{\varphi_{y}}$$
where φ and $\varphi_{y}$ are the curvature seismic demand and yield curvature, respectively. In this study, the $R_{\mathrm{cdd}}$ of the bottom cross-section is assessed, as the maximum moment and most severe damage are generated there. According to Equation (18), the $\varphi_{y}$ should be calculated first. Figure 16 shows the definition and time-dependent values of the yield curvature. As illustrated in Figure 16a, the equivalent bilinear method and equal area principle are adopted. Figure 16b indicates that the initial value of $\varphi_{y}$ is 0.00366 1/m and decreases continuously during the whole service life. After 100 years, the $\varphi_{y}$ is 0.0024 1/m and the corresponding decrease rate is 34.4%.

Figure 17 displays the time-histories of the drift ratio of the HRC bridge column under the excitation of ground motion 2 (GM2) and ground motion 8 (GM8). The time-history of the $R_{\mathrm{cdd}}$ of the bottom cross-section is also contained in the figure. To analyze the influence of chloride corrosion, the service times of 0, 50 and 100 years are selected as the comparison. It is seen that the peak and residual drift ratio response increases with the corrosion degree. The same phenomenon is observed for the curvature ductility demand ratio.

Using the selected suite of ground motions, 110 nonlinear time-history analyses are performed in total. For each analysis, the peak and residual drift ratio and *R*_cdd_ are summarized. The seismic resilience of structures has drawn much attention [51,52]. In the evaluation, the residual drift ratio is an important parameter. Figure 18 shows the distribution of HRC bridge column response over time. As shown in the figure, the seismic response of the structure exhibits some discreteness when subjected to different seismic excitation. Meanwhile, the gradual increases of the four parameters are demonstrated for all the bridge cases at 10-year time intervals. Compared with the sound structure, the median peak drift ratio and *R*_cdd_ grows by 131% and 251% after 100 years of service, respectively, while the corresponding increases for the median residual drift ratio and *R*_cdd_ are 183% and 1215%, respectively. The trend in the figure reveals that the deterioration induced by the marine environment enlarges the maximum seismic response of the HRC bridge column and results in more severe damage.

The displacement is chosen as the indicator to reflect the response of the abutment, and the peak displacement is adopted to quantify the seismic damage. Unlike from other components, abutment behavior in the longitudinal direction under the seismic excitation can be divided into active and passive behavior. The movement of the abutment towards the soil back-fill is passive behavior, while the pull away is active behavior. The passive resistance is partially provided by the soil and partially provided by the piles. For the active resistance, the contribution is mainly made by piles. The characteristic of the abutment results in different damage criteria for the two types of behaviors. Accordikng to Bayesian updated limit states for bridge components [53], the peak displacement ranges corresponding to the none and slight damage in the passive direction are [0, 37] and [37, 146] mm, respectively. As the moderate, extensive and complete damage are rarely observed in the actual project, they are not defined here (N/A). The peak displacement ranges corresponding to the five-level damages in the active direction are [0, 9.8], [9.8, 37.9], [37.9, 77.2], N/A and N/A, respectively.

Figure 19 displays the peak displacement responses over time for the two abutments of the HRC bridge. The damage distribution is also analyzed. As shown in the figure, the passive responses of the left abutment are larger than the active ones. The opposite phenomenon is observed for the right abutment. The peak displacements in the two directions are increased with the service time. Compared to the sound state, the increase of the median peak displacement of the left abutment in the passive and active direction after providing service for 100 years are 22.8% and 6.0%, respectively. The corresponding results for the right abutment are 5.2% and 21.6%, respectively. Considering the gradually increased seismic demand, the damage distribution at the service time of 100 years after 10 seismic excitations is analyzed and illustrated in the figure. The *D*_A_ and *D*_P_ refer to the displacement in the active and passive direction, respectively. It is seen that the abutment is a less vulnerable component in the bridge structure. The damage degree is mostly less than or equal to moderate damage.

The maximum deformation $\Delta_{\mathrm{def}}$ is adopted to quantify the seismic damage of the fixed and expansion bearings, and the time-dependent simulation results are shown in Figure 20. The damage state versus service time under the excitation of GM3, GM6 and GM9 is also depicted as the examples to exhibit the damage evolution. Moreover, the ranges of $\Delta_{\mathrm{def}}$ corresponding to the five damage levels of ND (no damage), SD (slight damage), MD (moderate damage), ED (extensive damage) and CD (complete damage) are given in the figure. It is seen that the deterioration of the FRP and steel bars would decrease the response of the fixed bearing. When subjected to GM3, the damage state changes from complete to slight damage during the whole service life. For the expansion bearings, the increasing trend of seismic response is not obvious, correspondingly, the damage state has a high probability of sustaining the same.

## 5. Conclusions

This paper studies the time-dependent seismic demand and seismic performance of the HRC bridge in the marine environment. The corrosion deterioration of reinforcement materials, bridge columns and the whole bridge structures is analyzed in three successive parts. The following key conclusions are reached based on the study results:(1)The cover depth significantly influences the corrosion initiation time of steel bars. Therefore, the decrease of diameter and yield strength is more severe for steel stirrups than the longitudinal steel bars. Main feature of corrosion deterioration of GFRP bars is reduction of tensile strength, which exhibits a relatively fast rate in the early 30 years and then gradually slows down.(2)The compression and tension behavior, as well as the fracture failure phenomenon of the GFRP bars, are critical for numerical simulation of the HRC columns and structures. The finite element modeling method proposed in this paper has a high precision in predicting structural seismic response.(3)The energy dissipation capacity, displacement ductility, bearing capacity and residual displacement of the HRC bridge column decrease with the service time. Furthermore, it is found when the hybrid ratio (area of GFRP bars/ area of steel bars) increases from 25.8% to 54.2% due to corrosion, the post-yield stiffness ratio of HRC columns increases from 0.174 to 0.240. In addition, compared to RC bridge columns, the bearing capacity of the HRC column increases by as much as 12.6%, and the residual displacement decreases by as much as 40.2%.(4)Under ground motion excitations, the peak and residual drift ratio/curvature ductility demand ratio of the HRC bridge columns have the positive correlation with the exposure time, indicating the gradually increasing damage. The abutment is a less vulnerable component in the HRC bridge structure, and the probability of less than extensive damage at 100 years is 90%. In the whole life cycle, the expansion bearing shows a 32% increase in maximum deformation, while the 76% decrease is obtained for the fixed bearing.(5)The study conducted in this paper is focused on the time-dependent performance deterioration of the coastal bridges induced by the chloride corrosion. The influence of fatigue damage, creep phenomenon and carbonization are not being considered. For future research, the seismic performance of the coastal structures under the action of multiple factors will be investigated.

To the best of the authors’ knowledge, this paper presents an early investigation on the time-dependent seismic performance of HRC bridge structures in the marine environment. Research results contribute to the widespread use of FRP materials in coastal regions.

## Figures and Tables

**Figure 1 materials-15-05293-f001:**
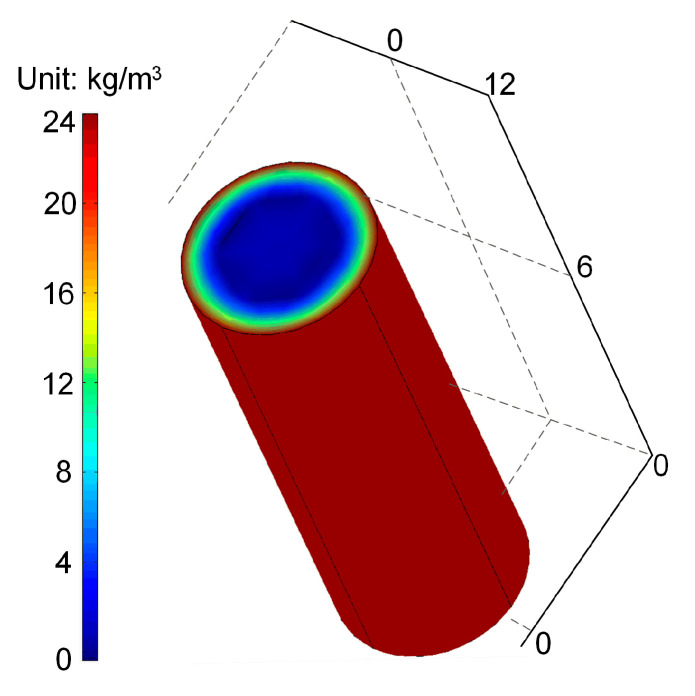
Simulation result of total chloride content distribution after servicing for 50 years.

**Figure 2 materials-15-05293-f002:**
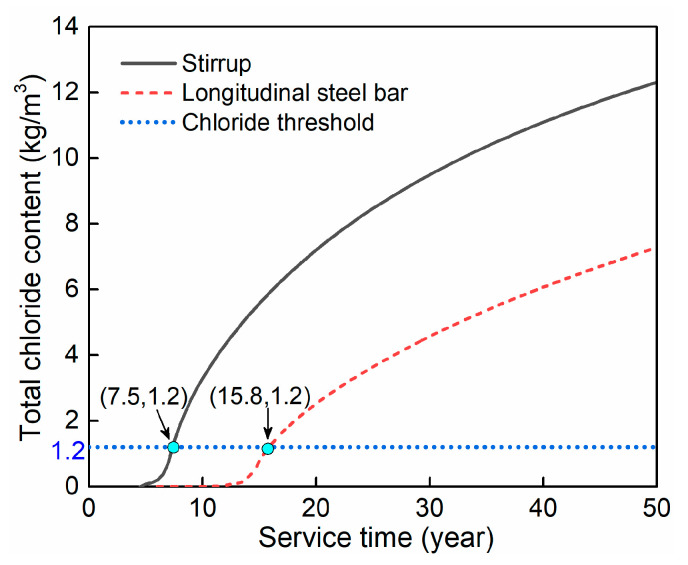
Corrosion initiation time of the longitudinal steel bar and stirrup.

**Figure 3 materials-15-05293-f003:**
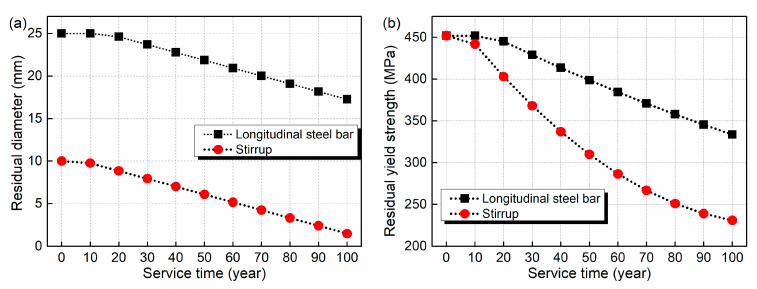
Estimated result of time-dependent steel bars performance influenced by chloride corrosion: (**a**) residual diameter; (**b**) residual yield strength.

**Figure 4 materials-15-05293-f004:**
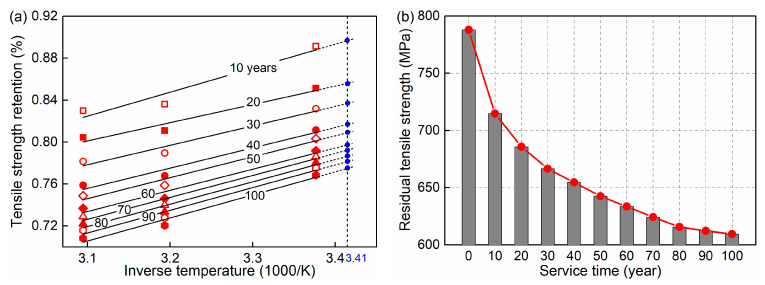
Estimated results of GFRP bar deterioration: (**a**) Arrhenius curves; (**b**) residual tensile strength.

**Figure 5 materials-15-05293-f005:**
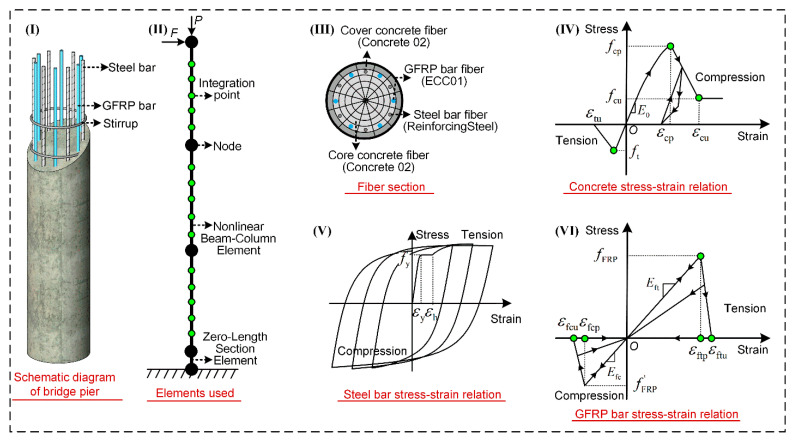
Finite element modeling method.

**Figure 6 materials-15-05293-f006:**
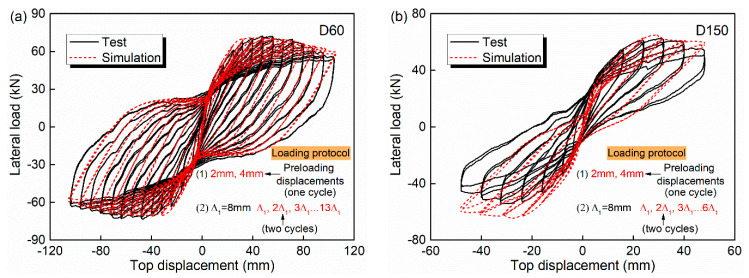
Simulation and test results of lateral load-top displacement: (**a**) Specimen D60; (**b**) Specimen D150.

**Figure 7 materials-15-05293-f007:**
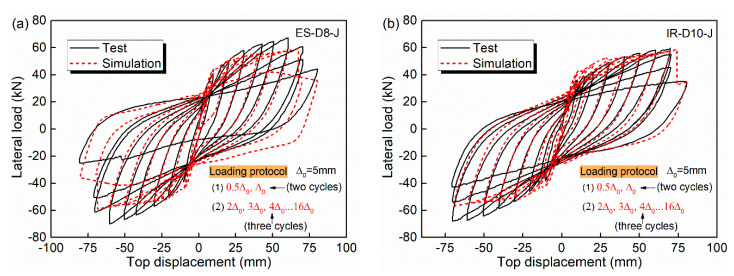
Simulation and test results of lateral load-top displacement: (**a**) Specimen ES-D8-J; (**b**) Specimen IR-D10-J.

**Figure 8 materials-15-05293-f008:**
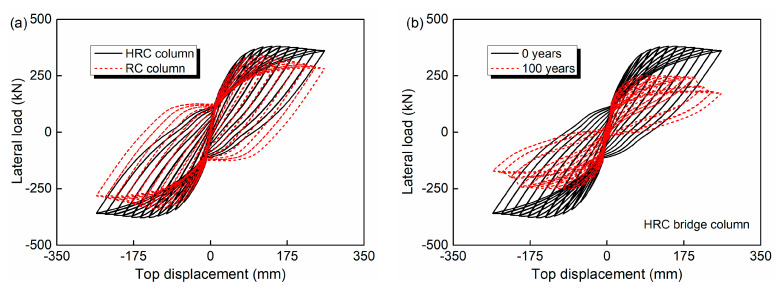
Lateral load-displacement curves: (**a**) HRC and RC bridge column at 0 years; (**b**) HRC bridge columns at 0 and 100 years.

**Figure 9 materials-15-05293-f009:**
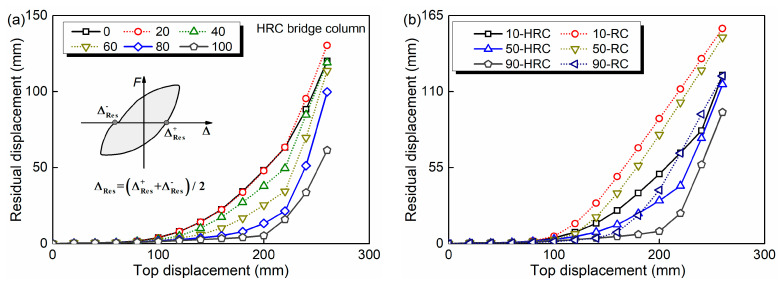
Time-dependent residual displacements of: (**a**) HRC bridge column; (**b**) HRC and RC bridge column.

**Figure 10 materials-15-05293-f010:**
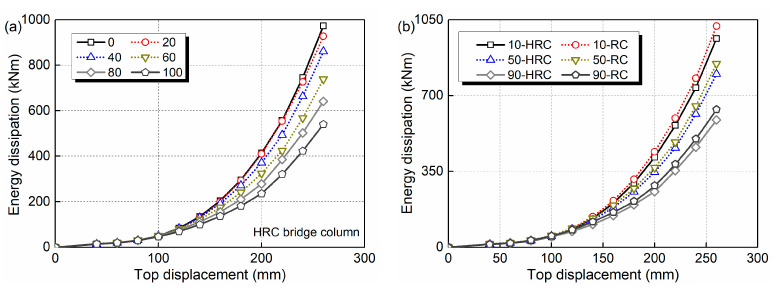
Time-dependent energy dissipations of: (**a**) HRC bridge column; (**b**) HRC and RC bridge column.

**Figure 11 materials-15-05293-f011:**
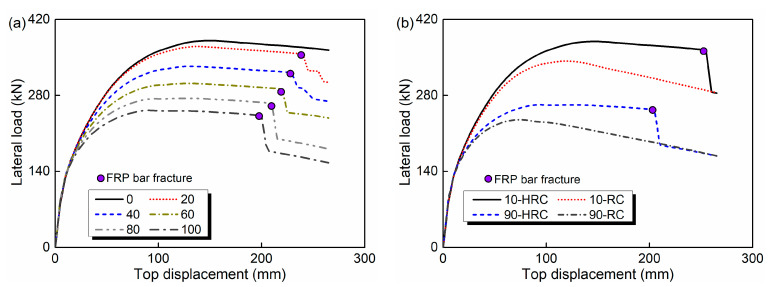
Time-dependent lateral load-top displacement envelope curves: (**a**) HRC bridge column; (**b**) HRC and RC bridge column.

**Figure 12 materials-15-05293-f012:**
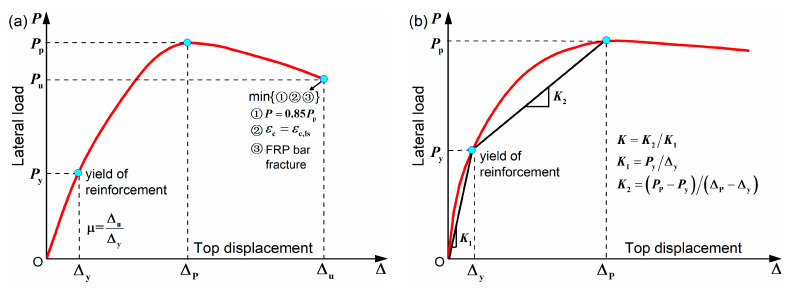
Definition of the characteristic parameters of envelope curves: (**a**) yield, peak, ultimate parameters and displacement ductility coefficient; (**b**) post-yield stiffness ratio.

**Figure 13 materials-15-05293-f013:**
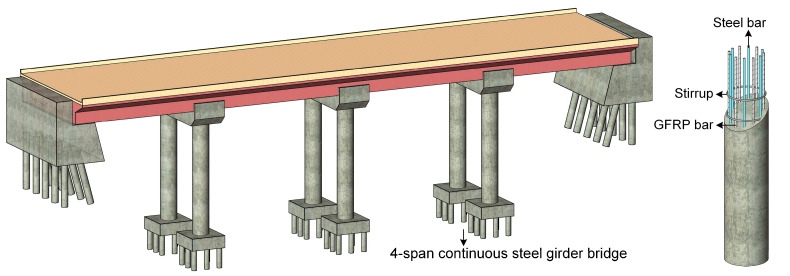
Schematic diagram of the analyzed HRC bridge.

**Figure 14 materials-15-05293-f014:**
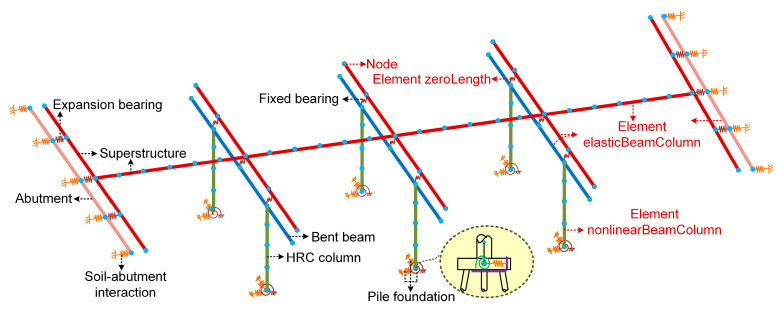
Finite element model of the analyzed HRC bridge.

**Figure 15 materials-15-05293-f015:**
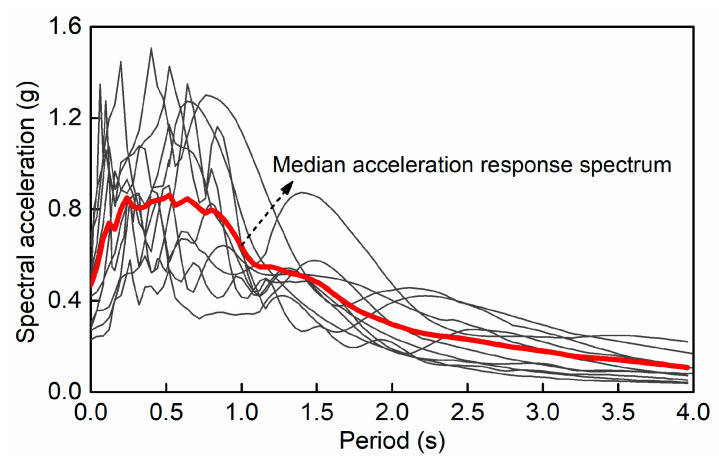
Acceleration response spectrum of the ground motions.

**Figure 16 materials-15-05293-f016:**
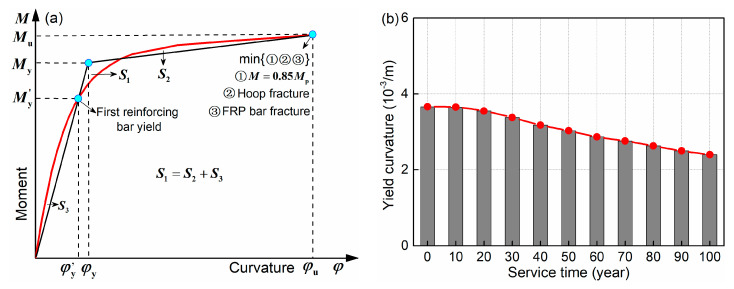
Yield curvature of the cross section: (**a**) definition; (**b**) time-dependent values.

**Figure 17 materials-15-05293-f017:**
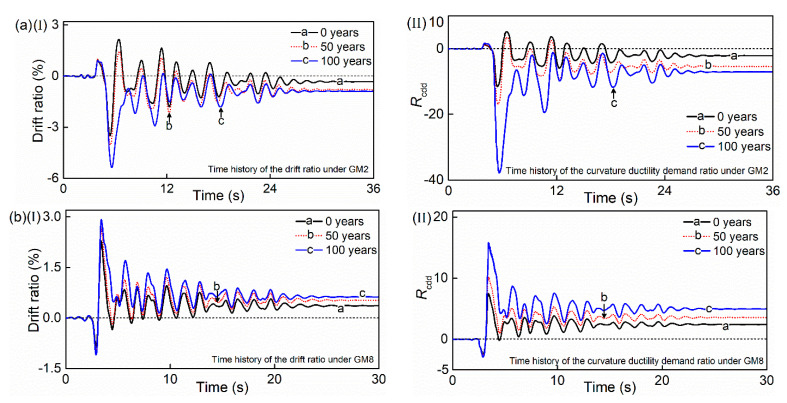
Time-dependent time histories of the drift ratio and curvature ductility demand ratio under the excitation of: (**a**) GM2; (**b**) GM8.

**Figure 18 materials-15-05293-f018:**
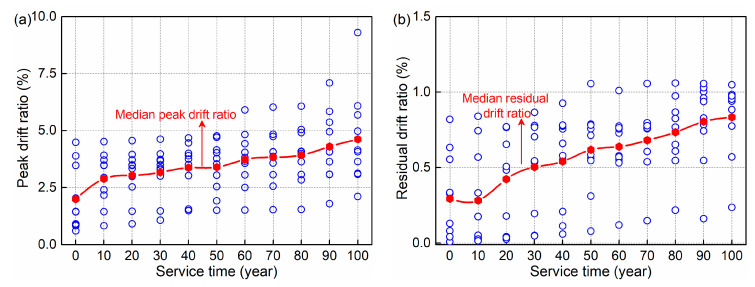
Distribution of HRC bridge column response over time: (**a**) peak drift ratio; (**b**) residual drift ratio; (**c**) peak *R*_cdd_; (**d**) residual *R*_cdd_.

**Figure 19 materials-15-05293-f019:**
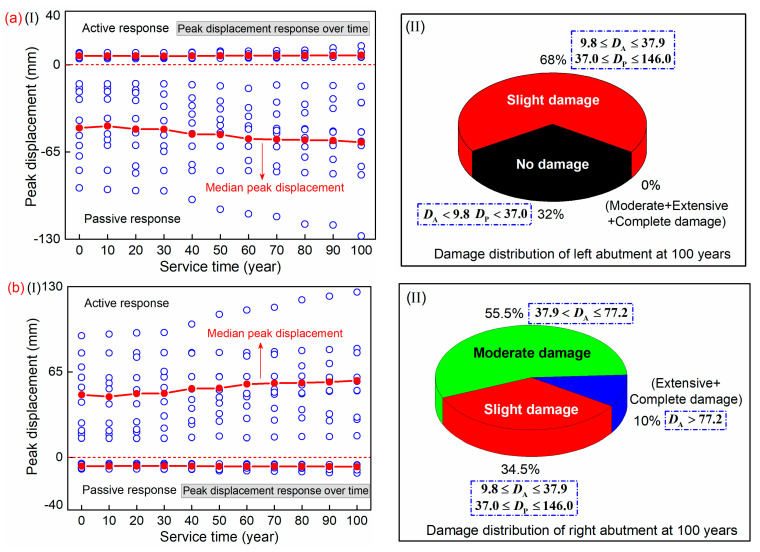
Peak displacement response over time and damage distribution of abutment: (**a**) left abutment; (**b**) right abutment.

**Figure 20 materials-15-05293-f020:**
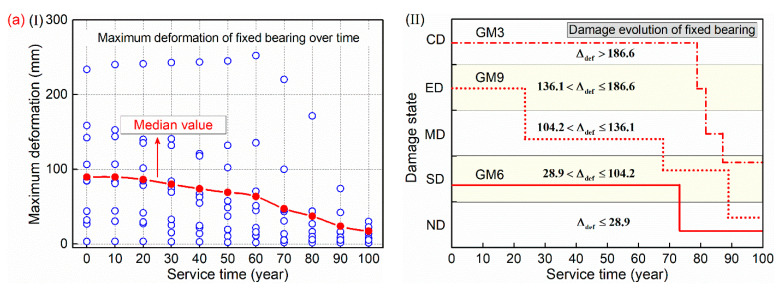
Maximum deformation and damage state over time: (**a**) fixed bearing; (**b**) expansion bearing.

**Table 1 materials-15-05293-t001:** Mathematical descriptions of the influence coefficients.

Function	Assigned Value	Meaning
$F_{1}(h)=\frac{1}{1+{\left[(1-h)/(1-h_{c})\right]}^{4}}$	$h_{c}$ = 0.75	$h_{c}$: critical humidity level
$F_{2}(T)=\exp\left[\frac{U}{R}(\frac{1}{T_{\mathrm{ref}}}-\frac{1}{T})\right]$	U = 41,800 (J/mol) $R=8.314\;(J/\mathrm{mol}.{}^{{}^{\circ}}K)$ $T_{\mathrm{ref}}=293\;({}^{{}^{\circ}}K)$	U: activation energy R: gas constant $T_{\mathrm{ref}}$: reference temperature
$F_{3}(t)={\left(\frac{t_{\mathrm{ref}}}{t}\right)}^{m}$	$t_{\mathrm{ref}}$ = 28 (day); m=0.35	$t_{\mathrm{ref}}$: reference time m: empirical age factor
$F_{4}(C_{f})=1-k_{\mathrm{ion}}{\left(C_{f}\right)}^{n}$	$k_{\mathrm{ion}}$ = 8.366; n=0.5	$k_{\mathrm{ion}}$, n: empirical coefficients

**Table 2 materials-15-05293-t002:** Comparisons between the simulation and test results.

Test Specimen	μ	Displacement Amplitude = 40 mm		Displacement Amplitude = 104 mm	
		ED(kN·m)	SS(kN/mm)	ED(kN·m)	SS(kN/mm)
D0	5.17/5.68 (9.86)	32.5/35.5 (9.23)	1.88/1.71 (9.04)	198/211 (6.57)	0.61/0.56 (8.20)
D30	4.51/5.11 (13.3)	31.9/34.2 (7.21)	1.87/1.71 (8.56)	194/210 (8.25)	0.60/0.54 (10.0)
D60	4.04/4.29 (6.19)	31.4/33.1 (5.41)	1.76/1.65 (6.25)	187/206 (10.2)	0.55/0.47 (14.5)
D105	3.63/3.52 (3.03)	27.3/24.1 (11.7)	1.51/1.57 (3.97)	SF	SF
D130	3.19/3.08 (3.45)	22.4/19.4 (13.4)	1.49/1.28 (14.1)	SF	SF
D150	3.17/2.91 (8.20)	19.6/15.5 (20.9)	1.48/1.24 (16.2)	SF	SF

**Table 3 materials-15-05293-t003:** Strength, ductility and stiffness of the HRC bridge columns with different service time.

Service Time (a)	$\Delta_{y}\;(\mathrm{mm})$	$\Delta_{P}(\mathrm{mm})$	$\Delta_{u}$ (mm)	$P_{y}\;(\mathrm{kN})$	$P_{p}\;(\mathrm{kN})$	$P_{u}\;(\mathrm{kN})$	$\mu_{\Delta }$	K
0	40.1	151	166	257	381	369	4.14	0.174
20	40.1	141	153	257	370	358	3.82	0.175
40	38.2	130	131	234	334	324	3.43	0.178
60	37.5	121	123	219	308	300	3.28	0.183
80	36.2	105	108	198	275	271	2.98	0.205
100	36.0	90	102	186	253	245	2.83	0.240

**Table 4 materials-15-05293-t004:** Ground motion records for dynamic time-history analysis.

No.	Event	Station	PGA (g)	$S_{a}(g)$
1	Superstition Hills, 1987	Parachute Test Site	0.432	0.890
2	Landers, 1992	Yermo Fire Station	0.245	0.498
3	Northridge, 1994	Newhall-W Pico Canyon Rd.	0.357	0.762
4	Cape Mendocino, 1992	Centerville Beach	0.318	0.412
5	Northridge, 1994	Pacoima Kagel Canyon	0.433	1.089
6	Parkfield-02_CA, 2004	Parkfield-Fault Zone 1	0.833	0.989
7	Northridge, 1994	Sylmar-Converter Sta	0.623	1.321
8	Parkfield-02 CA, 2004	Parkfield-Cholame 2WA	0.624	0.988
9	Christchurch, New Zealand, 2011	Pages Road Pumping Station	0.569	0.638
10	Loma Prieta, 1989	Foster City-APEEL 1	0.284	0.901

## Data Availability

Not applicable.

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
