# Peer review of "Time-Dependent Seismic Performance of Coastal Bridges Reinforced with Hybrid FRP and Steel Bars"

_materials, 2022, doi:10.3390/ma15155293_

Round 1

Reviewer 1 Report

The topic of the article is very interesting and in line with the goals and scope of the journal. However, I suggest a revision of the article in light of the comments that follow.

Page 1 - In the abstract, similar to the other acronyms, provide the description of "RC".

Page 2 - Although the objectives are well defined at the end of the introduction, the research questions that the results of this paper are intended to answer are not stated. The authors must clearly state how the contribution of this work differs from work that has already been done in this area.

Page 4 - Table 1. The second column shows a set of assigned values. How were these values determined? Are they based on a standard? Were they taken from other work? If applicable, provide the appropriate references. Clarify this question in the text of the article.

Page 4 - "Fig. 1 shows the distribution of total chloride content in the bridge column after servicing for 50 years." Figure 1 appears to represent a simulation, correct? If so, you should indicate in the text of the article and in the caption of Figure 1 that it is a simulation and that the values given are an estimate, i.e., not determined by experimental testing.

Page 6 - Figure 3: The variations in residual strength are due to corrosion, correct? You should include this information in the legend of Figure 3. How were these results obtained? Were they performed in the context of this article? If not, the appropriate references must be added. If they were obtained using the expressions given in the article, you should indicate that they are an estimate and not experimentally determined results.

Page 7 - Figure 4: Similar to the previous point, when using expressions to create these graphs, you should indicate that they are an estimate and not experimentally determined results.

Page 9 - "Model Validation": the authors validate the finite element model by comparing numerical and experimental hysteresis loops. However, it is not clear how the experimental results were obtained. Information about the experimental tests performed and the corresponding standards need to be added to the article so that they can be replicated. The number of hysteresis cycles performed per test piece must also be indicated in the text of the article and in the legends of Figures 5 and 7. It must also be indicated whether the hysteresis cycles correspond to the stabilized response or whether they were performed stepwise without waiting for the stabilized response of the material.

Page 10 - "Sismic performance analysis" It is not clear from the text how the authors address the problem of cumulative fatigue damage. Indeed, during the 100-year service life, there will be damage due to corrosion, but also due to cyclic loads of variable amplitude resulting from normal operation of the bridge, from seismic phenomena, and also from the creep phenomenon. The authors should clarify how they manage to separate these damages from each other, or how the aggregation of these damages with different causes is considered in this work.

Page 13 - Table 3, insert the appropriate units.

Page 14, "HRC bridge model", indicate the software used for the simulation and the simplifications assumed.

Page 20 - "Conclusions" - Conclusions should be updated with current work constraints and future improvements.

Reviewer 2 Report

This study deals with the time-dependent seismic performance of coastal bridges reinforced with hybrid FRP and steel bars. The content is suitable to publish on Materials. The manuscript is well written. The reviewer has some comments as follows:

1) This study performed the analysis on the deteriorated HRC bridges at three contents: (i) content of materials; (ii) content of columns; (iii) content of structures. The authors should make a connection between these contents.

2) Most of the results are from numerical simulations. To enhance the reliability of data, the results should be verified to the experimental results and/or the published results.

3) The authors need to explain why the Fick’s second law was chosen for the steel bar deterioration and the Arrhenius model was chosen for the FRP bar deterioration. Why weren’t other models chosen?

4) The values of P and F in Figure 5 must be clarified.

5) In Figure 5, the compression part of the steel bar must be added.

6) In Figure (8b), why is the top displacement between 0 year and 100 years similar? This is not reasonable. Please clarify.

7) In section 4, what was the software used? The type of elements, material properties, and boundary conditions of the FE model must be explained.

Round 2

Reviewer 1 Report

After analyzing the revised version of the article "Time-dependent seismic performance of coastal bridges reinforced with hybrid FRP and steel bars", it can be stated that the authors have significantly improved the article. In this sense, I believe that the article in the revised version meets the necessary requirements to be published in the journal Materials.

Sincerely,

Author Response

Comment:  After analyzing the revised version of the article "Time-dependent seismic performance of coastal bridges reinforced with hybrid FRP and steel bars", it can be stated that the authors have significantly improved the article. In this sense, I believe that the article in the revised version meets the necessary requirements to be published in the journal Materials.

Response: The authors wish to thank the reviewer for the very helpful suggestions.

Reviewer 2 Report

The authors have revised most of the reviewer’s comments. However, the reviewer still has two comments as follows:

1) This study performed the analysis on the deteriorated HRC bridges at three contents: (i) content of materials; (ii) content of columns; (iii) content of structures. The authors should make a connection between these contents (i.e., section 2, section 3, section 4). What data of section 2 was used for section 3? What data of section 2 and section 3 was used for section 4? Please clarify.

2) In Figure (8b), the displacements are over the ultimate displacements (Table 3) for both 0 year and 100 year. Please clarify.

Round 3

Reviewer 2 Report

The manuscript has been revised following the reviewer’s comments.

Author Response

The reviewer’s comment: "The manuscript has been revised following the reviewer’s comments".

Response: The authors wish to thank the reviewer for the valuable suggestions.